

# Adherence to the WHO recommendation of three weekly days of vigorous intensity activities in children: an accelerometry study of vigorous physical activity bouts

Juan Carlos Benavente-Marín[1], Francisco Javier Barón-López[1], Begoña Gil Barcenilla[2], Guadalupe Longo Abril[2], José M. Rumbao Aguirre[2], Napoleón Pérez-Farinós[1,3] and Julia Wärnberg[1,3]

[1] EpiPHAAN Research Group, Universidad de Málaga—Instituto de Investigación Biomédica de Málaga (IBIMA), Málaga, Spain
[2] Plan Integral de Obesidad Infantil de Andalucía (PIOBIN), Consejería de Salud y Consumo. Junta de Andalucía, Sevilla, Spain
[3] Centro de Investigación Biomédica en Red Fisiopatología de la Obesidad y la Nutrición (CIBEROBN), Instituto de Salud Carlos III, Madrid, Spain

Corresponding authors
Napoleón Pérez-Farinós,
napoleon.perez@uma.es
Julia Wärnberg, jwarnberg@uma.es

## ABSTRACT

**Background:** The World Health Organization (WHO) recommends that children and adolescents incorporate vigorous intensity activities (VIAs) at least three days a week. This recommendation has not been sufficiently studied using objective methods, such as accelerometry. Physical education classes and extracurricular sports activities are optimal opportunities for compliance with this recommendation.

**Objective:** To identify VIAs through bouts of vigorous physical activity (VPA-Bouts) evaluated with accelerometry and, with this, to know the compliance with the recommendation on VIAs.

**Methods:** A cross-sectional study of the habitual physical activity of 353 children (8–9 years old) was carried out using accelerometry and participation in organized extracurricular sports activities was asked through a questionnaire. School days with and without physical education class, weekends, and the average weekly day were identified, as well as school time and out-of-school time. A VPA-Bout was defined as an interval of at least 60 minutes with a proportion of VPA of at least 16.7% in boys and 12.5% in girls (10.0 and 7.5 minutes/hour of VPA, respectively).

**Results:** The average daily time in organized extracurricular sports activities declared by questionnaire and the average daily duration of the VPA-Bouts evaluated with accelerometers in the extracurricular period was 21.3 (SD 19.8) and 23.9 (SD 31.2) minutes, respectively, in boys, whereas, in girls it was 20.2 (SD 17.4) and 11.0 (SD 16.9) minutes, respectively. In school time including a physical education class, there was a higher proportion of VPA-Bouts than without these classes (with: 28.6%, without: 2.1%, $p < 0.001$). Children who reported at least three weekly hours of organized extracurricular sports activities accumulated a higher proportion of school afternoons with VPA-Bouts than those with fewer weekly hours of this type of activities (≥3 hours/week: 27.5%, <3 hours/week: 9.3%, $p < 0.001$). On the weekend, boys who reported at least three weekly hours of organized extracurricular sports activities performed more VPA-Bouts than those participating in less weekly hours, while in girls no significant differences were observed (weekend; boys,

≥3 hours/week: 26.0%, <3 hours/week: 9.0%, $p < 0.001$; girls: 8.3%, 8.0%, $p = 0.917$). Compliance with the recommendation to incorporate VIAs at least three days a week was 23.8%. Children who reported at least three weekly hours of extracurricular sports activities achieved higher compliance than those who reported fewer extracurricular activities (≥3 hours/week: 35.1%, <3 hours/week: 12.7%, $p < 0.001$). Additionally, boys showed higher compliance rates than girls (boys: 32.9%, girls: 15.3%, $p < 0.001$).

**Conclusion:** One in every four children met the WHO recommendation to incorporate VIAs at least three days a week, as evaluated by accelerometry. Physical education classes and extracurricular organized sports activities contributed to compliance with this recommendation.

# INTRODUCTION

The World Health Organization (WHO) recommends that children and adolescents should do at least an average of 60 minutes per day of moderate to vigorous, mostly aerobic, physical activity (MVPA) across the week and should incorporate vigorous-intensity aerobic activities (VIAs), as well as those that strengthen muscle and bone, on at least three days a week (*World Health Organization (WHO), 2020*; *Chaput et al., 2020*).

The VIAs could be defined as sessions of physical activity that incorporate enough vigorous-intensity physical activity (VPA), such as some sports activities, physical education classes, or active games. However, the WHO Guidelines on Physical Activity do not provide a specific definition of the VIAs duration or the minimum time of VPA they should contain. This situation hinders researchers using objective assessment tools, such as accelerometry, to calculate the prevalence of compliance with this recommendation or the effect of sessions of VIAs on health.

The WHO incorporates two distinct recommendations on physical activity, promoting different patterns of physical activity for health. The pattern of physical activity promoted by the recommendation on average daily MVPA has been, and continues to be, widely studied (*Guthold et al., 2020*; *Steene-Johannessen et al., 2020*). However, the pattern of physical activity that seems to be promoted by the recommendation on VIAs has not been properly studied using objective assessment methods. Therefore, the evaluation of VPA patterns concentrated in activity sessions using accelerometry is important for the study of the relationship between VIAs and child health (*Stone et al., 2009*; *Chinapaw et al., 2018*).

Accelerometry has proven to be a highly reliable and valid tool for assessing physical activity in children (*Sirard & Pate, 2001*; *Migueles et al., 2017*; *Gao et al., 2021*). Traditionally, this information had been collected through questionnaires, especially in larger population studies (*Guthold et al., 2020*; *Whiting et al., 2021*). However, the discrepancies found when trying to establish relationships between physical activity and

health outcomes in the child population using subjective methods (*Adamo et al., 2009*) have led to other objective assessment methods, such as accelerometry, gaining greater prominence in recent years.

Some research has demonstrated through accelerometry that physical education classes (*Meyer et al., 2013*; *Chen, Kim & Gao, 2014*; *Mooses et al., 2017*; *Calahorro-Cañada et al., 2017*; *Ikeda et al., 2022*; *Benavente-Marín et al., 2024*) and organized sports activities (*Silva et al., 2013*; *Marques, Ekelund & Sardinha, 2016*; *Ikeda et al., 2022*) contribute to the performance of more physical activity at all intensities, as well as to the achievement of the WHO's MVPA recommendation. However, none of these studies presented compliance with the recommendation to incorporate VIAs at least three days a week.

Participation in physical education classes and organized sports activities can be easily recorded through questionnaires (*Whiting et al., 2021*), but these activities do not always include VIAs and on certain occasions may not be sufficient to meet the recommendations on physical activity, especially in girls (*Kokko et al., 2019*; *Benavente-Marín et al., 2023*). Therefore, to enable the study of VIAs through accelerometry, we have previously proposed the use of VPA bouts (VPA-Bouts) derived from the VPA carried out in especially active physical education classes in children aged 8 to 9 years (*Benavente-Marín et al., 2023*). Using this methodology, the objectives of this study are: (1) To objectively identify VIAs in children during the week, both during school time and out-of-school time, and compare the daily frequency of VIAs according to participation in organized sports activities, (2) To calculate the prevalence of meeting the WHO's recommendation to incorporate VIAs at least three days a week.

# MATERIALS AND METHODS

## Study design and sample

We conducted a cross-sectional study to examine physical activity using accelerometry in a representative sample of 8- and 9-year-old children from the ALADINO 2019 Study (ALimentación, Actividad física, Desarrollo INfantil y Obesidad in Spain in 2019) in Andalusia, Spain. Andalusia is one of the 17 Autonomous Communities into which Spain is divided, located in the southern Iberian Peninsula. The ALADINO 2019 Study was conducted in Andalusia by the Spanish Agency for Food Safety and Nutrition (AESAN) in collaboration with the Andalusian regional childhood obesity plan PIOBIN (Plan Integral de Obesidad Infantil de Andalucía). In the ALADINO 2019 study, 40 primary education schools in Andalusia participated. Both the design and methodology of the ALADINO 2019 Study were developed in accordance with the protocols and recommendations of the WHO European Childhood Obesity Surveillance Initiative (COSI Euro WHO) (*ALADINO, 2020*; "COSI Euro WHO").

A pre-determination of the minimum sample size necessary for each of the estimates to be made in the study was calculated. The one that required the largest sample size was the estimation of the 95% confidence interval for the calculation of the proportion of compliance with the WHO's VIAs recommendation. The expected proportion was set at 0.25 and it was decided that the total width of the confidence interval would be 0.1. As a result, it was determined that the sample size should be at least 307 subjects.

An average classroom size of 20 students in 3$^{rd}$ grade of primary education was assumed, with a 20% refusal rate to participate. Therefore, to reach the estimated sample size, we needed to evaluate one 3rd-grade classroom in 19 primary education schools. For this current study, all 40 primary education schools that participated in the ALADINO 2019 Study in Andalusia were invited to participate, thereby reaching and surpassing the calculated minimum sample size.

Inclusion criteria for this study were: (1) enrollment in the 3rd grade of primary education during the 2019/2020 academic year in a primary education school participating in the ALADINO 2019 Study in Andalusia; and (2) having a signed informed consent from legal guardians authorizing participation in the specific accelerometry study. Participants with limitations for physical activity during the evaluation and those over 9.99 years old were excluded from the analysis.

We reported this study as per the Strengthening the Reporting of Observational Studies in Epidemiology (STROBE) guideline (Supplemental STROBE Statement) (*Cuschieri, 2019*). This study is a supplementary study to the ALADINO 2019 study in Andalusia, incorporating newly collected accelerometer data. The legal guardians of the participants were offered the opportunity to obtain a report of their children's individual results. Similarly, participating schools were offered the opportunity to obtain a report of the average results of the participating student group.

The study was conducted in accordance with the Declaration of Helsinki (*World Medical Association, 2013*) and approved by the research ethics committee CEI -Costa del Sol and the Portal de Ética de la Investigación Biomédica de Andalucía-PEIBA, the 26$^{th}$ of September 2019, with the reference number 0114-2019. All data was carried out respecting the European legislation 2016/679 of data protection, and the Spanish 'Organic Law 3/2018 of December 2005'. The clinical data was kept segregated and encrypted. The signed informed consent was obtained from all legal guardians.

## Data collection procedure

In the schools that agreed to participate, all 3$^{rd}$ grade primary school children from the classroom selected for the ALADINO 2019 Study in Andalusia were invited. The legal guardians of the children received the invitation in written form with all the information about the study.

Data collection was distributed evenly during the 2019–20 school year according to the size of the populations where the participating schools are located. Data collection was planned to be carried out between October 2019 and June 2020, but it ended in March 2020 due to the pandemic caused by the SARS-CoV-2 virus. In Spain, schools were closed from March 15, 2020, until the end of the 2019–20 academic year and home confinement of the population was decreed. For this reason, the results of this study are prior to the COVID-19 pandemic.

Two visits were made to each school. In the first visit, accelerometers were individually placed on the participants. The teachers and participants were asked to maintain their daily activity during the accelerometry evaluation. In the second visit, the accelerometers were removed and each participant was asked if they had removed the accelerometer or if

they had missed class during the evaluation. The schedule of the school time and physical education classes on the evaluated days was recorded. All participating schools began at 9:00 AM and ended at 2:00 PM. All had a daily recess of 30 minutes around the middle of the school time, starting between 11:00 AM and 12:00 PM.

## Criteria for accelerometry data collection

For the objective evaluation of physical activity, GENEActiv accelerometers (Activinsights Ltd., Kimbolton, UK) were used. These are triaxial accelerometers with a dynamic range of ±8 gravitational units (g), where 1 g equals Earth's gravitational pull. The accelerometers were configured with a sampling frequency of 40 Hz using GENEActiv PC Software (version 3.2).

The accelerometers were worn on the non-dominant wrist, and participants were asked to wear them continuously for at least 8 consecutive days to ensure a complete assessment of five school days and a weekend. Participants and their families were instructed not to remove the device at any time during the assessment (24 h protocol). It was emphasized that the device was waterproof, and participants were required to wear it while sleeping.

## Processing of accelerometry data

No noise filter was applied prior to processing. Raw accelerometer data files were processed using R (http://cran.r-project.org) with the R package accelerator (version 0.4.0) (*Barón-Suárez et al., 2023*). The processing included the processing functions of the R GGIR package (version 2.9.2) (*Migueles et al., 2019*). In summary, GGIR performed the following tasks: (1) Auto-calibration (*van Hees et al., 2014*); (2) Detection of abnormally high sustained values; (3) Non-wear time detection; (4) Calculation of the Euclidean norm minus one with negative values set to zero (ENMONZ or ENMO) (*van Hees et al., 2013*). The raw data were simplified by calculating ENMONZ values (measured in milligravitational units, mg) in 5-second epochs (*Baquet et al., 2007*; *Aadland et al., 2018*).

The GGIR algorithm was found to be inadequate in detecting relatively short non-wear periods, so the GGIR non-wear time definition was supplemented with strict periods of sustained inactivity. These periods needed to last at least 30 minutes, with angle changes in the Z-axis below two degrees, calculated between 8:00 AM and 10:00 PM.

To classify physical activity by intensity, the cut-off points published by *Hildebrand et al. (2014)*, *(2017)* for GENEActiv accelerometers, placed on the non-dominant wrist, in children aged 7 to 11 years, and expressed in ENMONZ (mg) were used. The specific cut-off points used were as follows: light physical activity (LPA, from 56.3 to 191.6 mg), moderate physical activity (MPA, from 191.6 to 695.8 mg), VPA (over 695.8 mg), physical activity at any intensity (LMVPA, over 56.3 mg), and MVPA (over 191.6 mg).

When participants reported removing the accelerometer for a known sporting activity, it was checked if this coincided with non-wear time. If confirmed, non-wear time was replaced with mean values for a similar sporting activity, which had been observed and studied in other participants from the same sample.

Five types of day and two daily segments were defined for the analysis. The types of day were: weekly day, school day, school day with physical education class, school day without

physical education class, and weekend day; while the daily segments were: from getting out of the bed until 2:00 PM (Before 2:00 PM), and from 2:00 PM until going to bed for the night's rest (After 2:00 PM). Awake time was also defined as the time between getting out of the bed and going to bed for the night's rest, and out-of-school time as the time spent on school days in the segment After 2:00 PM and the time spent on the whole day on the weekend.

An evaluated day was considered valid when the accelerometer was active and recording for a minimum of 20 hours (from 00:00 to 00:00 hour) with no more than two hours of non-wear time accumulated between 8:00 AM and 10:00 PM. Awake time was valid if it did not accumulate more than two hours of non-wear time, while a daily segments Before or After 2:00 PM was valid if it did not accumulate more than one hour of non-wear time each one. School time was valid if the accelerometer recorded at least four hours with no more than one hour of non-wear time during school hours. A physical education class was considered valid if it accumulated less than one minute of non-wear time, included at least three minutes of MVPA, or did not exceed 30 minutes of sedentary behavior (*i.e.*, epochs with less than 56.3 mg). Physical education classes for non-participating children were excluded.

Consequently, a valid school day implied a valid school time. If a valid school day also included a valid physical education class, it was considered a school day with physical education class. If the valid day was a Saturday or Sunday, it was considered a weekend day. To calculate the average weekly day, daily average results were weighted with 5/7 for the average of school days and 2/7 for the average of weekend days. Similarly, for out-of-school time, the results of the segment After 2:00 PM on school days (5/7) and the whole day of the weekend (2/7) were weighted. If an assessment had two identical days of the week (*e.g.*, two Mondays), these were averaged, and this average was used as the mean value for that type of day.

An assessment was considered valid when it had at least four valid weekly days (*Antczak et al., 2021*), of which at least one day was a school day with a physical education class, at least another school day without a physical education class, and at least another day was a weekend. In addition, for the assessment to be valid, it had to have at least one segment Before and another After 2:00 PM on each type of day analyzed. To maintain precision, holidays and school absence days were excluded from the analysis.

If there was non-wear time in the resulting valid days or segments, it was imputed by the average value of the different intensities of physical activity calculated for that same type of day or daily segment in the time interval occupied by the non-wear time. If the average value for that same type of day or segment was not available, it was imputed by the average weekly daily value for that time interval.

## Vigorous intensity activities and compliance with WHO recommendation

VIAs were operationalized through VPA-Bouts, that is, bouts of at least 60 minutes duration with at least 16.7% in boys and 12.5% in girls of VPA (*i.e.*, 10.0 and 7.5 minutes/hour of VPA). The VPA-Bouts were previously established in this same population (*Benavente-Marín et al., 2023*).

The prevalence of compliance with the WHO recommendation to "include VIAs at least three days a week" (*World Health Organization (WHO), 2020*), was calculated considering those participants who had at least one VPA-Bout in at least 3/7 valid weekly days as compliant.

## Other study variables

Information regarding sex and date of birth was collected in the informed consents. Age was calculated as the difference between the start date of the accelerometry evaluation and the participant's date of birth.

Body weight and height were measured between October and December 2019. The TANITA model UM-076 scale was used, capable of recording weights from zero to 150 kg with a precision of 100 g, and the portable SECA model 206 stadiometer, which measures between zero and 220 cm with a precision of 1 mm. Body Mass Index (BMI) was calculated as weight divided by height squared (kg/m$^2$). Weight status was classified into three categories (normal weight, overweight, and obesity) using the growth standards of the WHO (*de Onis et al., 2007*).

Following the COSI Euro WHO protocol, the legal guardians of the participants were asked the following question through a self-administered questionnaire: "Is your child a member of one or more sports or dance clubs (*e.g.*, football, athletics, hockey, swimming, tennis, basketball, judo, taekwondo, gymnastics, ballet, physical training, ballroom dancing, *etc.*,) or does he/she take classes in them?" The possible answers were "Yes" or "No". In case of an affirmative answer, the following question was asked: "In a normal week (including the weekend), how many hours does your child spend in these types of sports and physical activities?" The possible answers ranged from zero to 11 hours per week (*Whiting et al., 2021*).

It was considered that the children whose legal guardians responded that they did not participate in these types of sports and physical activities (*i.e.*, extracurricular organized sports activities) performed zero weekly hours of extracurricular organized sports activities. The children were classified into three groups: those who reported doing zero weekly hours; those who reported between one and two weekly hours; and those who reported at least three weekly hours of extracurricular organized sports activities. In addition, the first two groups were combined into one, leaving the following two groups: those who reported doing less than three weekly hours and those who reported at least three weekly hours.

## Statistical analysis

The mean, standard deviation (SD), minimum, maximum, and total valid days were calculated for the five types of day (*i.e.*, weekly day, school day, school day with physical education class, school day without physical education class, and weekend day) and daily segments studied (*i.e.*, Before 2:00 PM and After 2:00 PM). The average time of LPA, MPA, VPA, as well as the VPA performed within the VPA-Bouts, were calculated for the participants in all types of day and daily segments studied. The average daily duration of weekly day, awake time, Before 2:00 PM, After 2:00 PM, VPA-Bouts, and out-of-school

time VPA-Bouts and self-reported extracurricular organized sport activities was also calculated.

A description of the study sample was conducted: for quantitative variables, the mean and SD were calculated, and for qualitative variables, frequency and proportion were determined. To assess sex differences in all studied variables, the chi-square test was employed for qualitative variables, and the Student's t-test was used for quantitative variables if they followed a normal distribution, or the Mann-Whitney U test in case of non-normality.

The average percentage and SD of days and daily segments with at least one VPA-Bout were calculated for all types of day studied, by sex as well as for participants segmented between those who reported doing less than three weekly hours and at least three weekly hours of organized sports activities. If a VPA-Bout extended across the two daily segments, both segments were counted as a segment with a VPA-Bout.

To evaluate whether there were differences between the percentage of segments Before and After 2:00 PM, the Wilcoxon signed-rank test was used, while to know the differences between those who reported doing less than three weekly hours and those who reported at least three weekly hours of organized sports activities, the Mann-Whitney U test was used. The Wilcoxon signed-rank test was also used to evaluate whether there were differences between different types of day (*i.e.*, school days, with and without physical education class, and weekend) or segments (*i.e.*, After 2:00 PM on school days and whole day on weekends).

The proportion of children who met the WHO recommendation of "incorporating VIAs at least three days a week" was calculated. The chi-square test was used to assess whether there were differences between categories.

For all analyses, a significance level of $p < 0.05$ was established. The statistical analysis was performed with the SPSS software (IBM® SPSS® Statistics) version 25 for macOS (IBM Software Group, Chicago, IL).

## RESULTS

A total of 33 schools agreed to participate in the present study, out of the 40 schools invited. A total of 22 groups of 3rd grade primary school children were evaluated between October 2019 and March 2020. The remaining 11 groups of children were not evaluated due to the closure of schools caused by the COVID-19 pandemic. A total of 510 informed consents were delivered in the 22 evaluated schools (an average of 23.2 children per classroom). A total of 401 (78.6%) were correctly filled out and returned, of which, 385 children agreed to participate (75.5% of the total invited). Seven of the children authorized to participate were absent from class on the day the evaluation began and could not be evaluated, and in two evaluations, the data recorded in the accelerometer could not be extracted. No participant was excluded for being over 9 years old or having limitations for physical activity practice during the evaluation. Therefore, the sample that had data derived from the accelerometer evaluation was 376 children (73.7% of the total invited), of which, 353 children had at least four valid days, with at least one school day with a physical education class, one school day without a physical education class, and one weekend day,
as well as, at least one segment Before 2:00 PM and another After 2:00 PM for each of the types of day studied.

Table 1 shows the descriptive results of the types of day, as well as the segments Before and After 2:00 PM for the analyzed sample.

Table 2 shows the descriptive results in the total of participants, and by sex. No statistically significant differences were found between boys and girls in anthropometric data or in the prevalence of overweight and obesity.

Table 3 shows the average duration of the day, awake time, and the segment Before and After 2:00 PM, as well as the physical activity performed at different intensities (including VPA performed within the VPA-Bouts) on the average weekly day. It also shows the average daily duration of the VPA-Bouts in the whole weekly day and in the out-of-school time, as well as the average declared duration of organized extracurricular sports activities. The average daily times of VPA in VPA-Bouts, VPA, MVPA, and MPA were significantly higher in boys than in girls. The average daily time of LPA was significantly higher in girls than in boys. In the out-of-school time, VPA-Bouts were identified with an average daily duration of 17.2 minutes (SD 25.6), and the average daily duration of extracurricular sports activities declared by questionnaire was 20.8 minutes (SD 18.5).

Table 4 shows the average percentage of different types of day and daily segments where at least one VPA-Bout was identified. The results are shown for all participants, as well as segmented by sex. Participants performed on average at least one VPA-Bout on 23.8% (SD 23.1) of the average weekly days. On school time with a physical education class, the average number of school time with a VPA-Bout was 28.6% (SD 36.0), while on school time without a physical education class it was 2.1% (SD 9.6). In extracurricular time, no significant differences were found between the segment of after school time with and without a physical education class (18.9% *vs* 17.6%; $p = 0.417$), while there were differences between the afternoons of school days and whole weekend days (17.8% *vs* 13.4%; $p = 0.004$). The school day with physical education class was the only type of day where a higher proportion of VPA-Bout was observed in the school time compared to out-of-school time (28.6% *vs* 18.9%; $p < 0.001$).

Table 5 shows the average percentage of days and daily segments with VPA-Bout in boys and girls according to their participation in organized extracurricular sports activities. In boys, significant differences were observed in all segments of out-of-school time between those who declared not to perform or perform less than three weekly hours of extracurricular organized sports activities and those who declared to perform at least three weekly hours (After school time with physical education class: 8.2% *vs* 43.4%, $p < 0.001$; After school time without physical education class: 15.2% *vs* 30.8%, $p = 0.001$; Weekend day: 9.0% *vs* 26%, $p < 0.001$), while in girls no differences were observed on weekend days (After school time with physical education class: 8.2% *vs* 18.8%, $p = 0.006$; After school time without physical education class: 6.7% *vs* 20.2%, $p < 0.001$; Weekend day: 8.0% *vs* 8.3%, $p = 0.917$). Therefore, in the total participants, a higher proportion of school afternoons with VPA-Bouts was observed among those children who declared to perform at least three hours of organized extracurricular sports activities ($\geq$3 hours/week: 27.5%, <3 hours/week: 9.3%, $p < 0.001$). However, in the proportion of school times with VPA-

**Table 1 Descriptive statistics of types of day and daily segments.**

| Type of day/segment | Participants | Minimum | Maximum | Total days | Mean | SD |
|---|---|---|---|---|---|---|
| **Weekly days** | 353 | 4 | 9 | 2,469 | 7.0 | 1.1 |
| Before 2:00 PM | 353 | 4 | 9 | 2,483 | 7.0 | 1.1 |
| After 2:00 PM | 353 | 4 | 9 | 2,464 | 7.0 | 1.1 |
| **School days** | 353 | 2 | 5 | 1,632 | 4.6 | 0.6 |
| Before 2:00 PM | 353 | 2 | 5 | 1,645 | 4.7 | 0.5 |
| After 2:00 PM | 353 | 2 | 5 | 1,629 | 4.6 | 0.6 |
| **School days with PEC** | 353 | 1 | 3 | 681 | 1.9 | 0.7 |
| Before 2:00 PM | 353 | 1 | 3 | 684 | 1.9 | 0.7 |
| After 2:00 PM | 353 | 1 | 3 | 690 | 2.0 | 0.7 |
| **School days without PEC** | 353 | 1 | 4 | 952 | 2.7 | 0.7 |
| Before 2:00 PM | 353 | 1 | 4 | 962 | 2.7 | 0.7 |
| After 2:00 PM | 353 | 1 | 4 | 952 | 2.7 | 0.7 |
| **Weekend days** | 353 | 1 | 4 | 837 | 2.4 | 0.8 |
| Before 2:00 PM | 353 | 1 | 4 | 838 | 2.4 | 0.8 |
| After 2:00 PM | 353 | 1 | 4 | 835 | 2.4 | 0.8 |
| **Holiday days** | 125 | 1 | 2 | 229 | 1.8 | 0.4 |
| **School absence days** | 50 | 1 | 4 | 59 | 1.2 | 0.6 |

Note:
SD, standard deviation; PEC, physical education class.

**Table 2 Descriptive statistics of study participants.**

| | All participants | | | Boys | | | Girls | | | |
|---|---|---|---|---|---|---|---|---|---|---|
| | n | Mean | SD | n | Mean | SD | n | Mean | SD | p |
| **Age (years)** | 353 | 8.5 | 0.4 | 170 | 8.5 | 0.4 | 183 | 8.6 | 0.4 | 0.446 |
| **Weight (kg)** | 320 | 32.1 | 8.5 | 151 | 32.6 | 8.7 | 169 | 31.6 | 8.4 | 0.229 |
| **Height (cm)** | 320 | 131.4 | 6.4 | 151 | 132.0 | 6.4 | 169 | 130.9 | 6.3 | 0.129 |
| **BMI (kg/m$^2$)** | 320 | 18.4 | 3.6 | 151 | 18.5 | 3.7 | 169 | 18.3 | 3.6 | 0.419 |
| | n | % | | n | % | | n | % | | p |
| **Weight status** | | | | | | | | | | |
| Normal weight | 159 | 49.7 | | 70 | 46.4 | | 89 | 52.7 | | 0.413 |
| Overweight | 78 | 24.4 | | 37 | 24.5 | | 41 | 24.2 | | |
| Obesity | 83 | 25.9 | | 44 | 29.1 | | 39 | 23.1 | | |

Note:
n, number of participants; SD, standard deviation; p, p-value for the difference between boys and girls (Mann-Whitney U test or chi-square test); kg, kilograms; cm, centimeters; m, meters; BMI, body mass index.

Bouts, no significant differences were observed according to the declared weekly duration of extracurricular sports activities (≥3 hours/week: 14.0%, <3 hours/week: 12.7%, $p = 0.477$).

Table 6 shows the percentage of compliance with the WHO recommendation to "incorporate VIAs at least three days a week" evaluated using accelerometry. 23.8% of the children accumulated at least one VPA-Bout in at least three of seven average weekly days.

**Table 3 Average day and segments durations and average weekly physical activity, by sex.**

| | All participants | | | Boys | | | Girls | | | |
|---|---|---|---|---|---|---|---|---|---|---|
| | *n* | Mean | SD | *n* | Mean | SD | *n* | Mean | SD | *p* |
| **Weekly wear time (h)** | | | | | | | | | | |
| Whole day | 353 | 24.0 | 0.3 | 170 | 23.9 | 0.3 | 183 | 24.0 | 0.4 | 0.089 |
| Awake time | 353 | 14.7 | 0.5 | 170 | 14.7 | 0.6 | 183 | 14.6 | 0.5 | 0.340 |
| Before 2:00 PM | 353 | 5.7 | 0.5 | 170 | 5.7 | 0.5 | 183 | 5.7 | 0.5 | 0.846 |
| After 2:00 PM | 353 | 8.9 | 0.7 | 170 | 9.0 | 0.7 | 183 | 8.9 | 0.7 | 0.266 |
| **Weekly PA (min/day)** | | | | | | | | | | |
| VPA in VPA-Bouts | 353 | 4.7 | 6.4 | 170 | 7.0 | 7.9 | 183 | 2.7 | 3.5 | <0.001 |
| VPA | 353 | 15.5 | 9.2 | 170 | 20.0 | 10.5 | 183 | 11.4 | 5.1 | <0.001 |
| MPA | 353 | 70.6 | 21.3 | 170 | 76.8 | 22.9 | 183 | 64.9 | 17.9 | <0.001 |
| LPA | 353 | 218.3 | 33.4 | 170 | 212.4 | 33.3 | 183 | 223.8 | 32.7 | 0.001 |
| MVPA | 353 | 86.5 | 28.9 | 170 | 97.3 | 31.6 | 183 | 76.4 | 21.8 | <0.001 |
| LMVPA | 353 | 304.8 | 52.2 | 170 | 309.7 | 55.3 | 183 | 300.3 | 48.9 | 0.139 |
| **VPA-Bouts (min/day)** | | | | | | | | | | |
| Whole weekly day | 353 | 23.8 | 23.1 | 170 | 29.9 | 25.5 | 183 | 18.2 | 19.0 | <0.001 |
| Out of school | 353 | 17.2 | 25.6 | 170 | 23.9 | 31.2 | 183 | 11.0 | 16.9 | <0.001 |
| **Self-report OSA (min/day)** | 320 | 20.8 | 18.5 | 151 | 21.3 | 19.8 | 169 | 20.2 | 17.4 | 0.782 |
| | *n* | % | | *n* | % | | *n* | % | | *p* |
| **Self-report OSA** | | | | | | | | | | |
| 0 h/week | 102 | 31.9 | | 51 | 33.8 | | 51 | 30.2 | | 0.221 |
| 1–2 h/week | 64 | 20.0 | | 24 | 15.9 | | 40 | 23.7 | | |
| ≥3 h/week | 154 | 48.1 | | 76 | 50.3 | | 78 | 46.1 | | |

**Note:**
*n*, number of participants; SD, standard deviation; *p*, *p*-value for the difference between boys and girls (Mann-Whitney U test, t-test or chi-square test); h, hour; min, minutes; OSA, extracurricular organized sport activities; VPA-Bouts, bouts of ≥60 min with ≥16.7% in boys or 12.5% in girls of VPA; PA, physical activity; V, vigorous; M, moderate; L, light.

The children who declared to perform at least three weekly hours compared to those who declared to perform less than three weekly hours of organized extracurricular sports activities, showed a significantly higher proportion of compliance with the WHO's VIAs recommendation (≥3 hours/week: 35.1%; <3 hours/week: 12.7%; *p* < 0.001).

# DISCUSSION

This study shows the percentage of compliance with the WHO recommendation to "incorporate vigorous intensity activities (VIAs) at least three days a week" (*World Health Organization (WHO), 2020*), through a new method based on the processing of accelerometry data. With this method, the presence of physical education classes and extracurricular organized sports and physical activities, that are sufficiently active, has been identified. Participation in physical education classes and in extracurricular sports and physical activities contributed to the performance of VIAs, and thereby, to the compliance with the WHO recommendation.

A total of 23.8% of the evaluated children met the recommendation to "incorporate VIAs at least three days a week". In contrast, 80.6% of this same population met the WHO

Table 4 Average percentage of days and daily segments with bouts of vigorous activities on different types of day by sex.

| | All (n = 353) | | | Boys (n = 170) | | | Girls (n = 183) | | | |
|---|---|---|---|---|---|---|---|---|---|---|
| | % | SD | $p^*$ | % | SD | $p^*$ | % | SD | $p^*$ | $p^\dagger$ |
| **Weekly day** | 23.8 | 23.1 | | 29.9 | 25.5 | | 18.2 | 19.0 | | <0.001 |
| Before 2:00 PM | 10.7 | 12.9 | <0.001 | 13.0 | 13.7 | <0.001 | 8.5 | 11.8 | 0.106 | 0.001 |
| After 2:00 PM | 15.8 | 20.2 | | 21.0 | 23.6 | | 11.1 | 15.1 | | <0.001 |
| **School day with PEC** | 40.8 | 40.8 | | 48.9 | 41.3 | | 33.3 | 38.9 | | <0.001 |
| Before 2:00 PM (1) | 28.6 | 36.0 | <0.001 | 33.0 | 36.7 | 0.072 | 24.4 | 34.8 | <0.001 | 0.017 |
| After 2:00 PM (2) | 18.9 | 31.9 | | 25.7 | 37.6 | | 12.6 | 23.9 | | 0.002 |
| **School day without PEC** | 19.3 | 30.1 | | 24.9 | 32.4 | | 14.0 | 26.7 | | <0.001 |
| Before 2:00 PM (3) | 2.1 | 9.6 | <0.001 | 3.6 | 12.4 | <0.001 | 0.7 | 5.7 | <0.001 | 0.004 |
| After 2:00 PM (4) | 17.6 | 28.8 | | 22.5 | 31.5 | | 13.0 | 25.3 | | 0.001 |
| **School day** | 28.0 | 27.2 | | 34.5 | 28.8 | | 21.9 | 24.2 | | <0.001 |
| Before 2:00 PM | 13.1 | 16.4 | 0.001 | 15.6 | 17.1 | 0.001 | 10.7 | 15.4 | 0.405 | 0.005 |
| After 2:00 PM (5) | 17.8 | 24.2 | | 23.4 | 27.7 | | 12.7 | 19.1 | | <0.001 |
| **Weekend day (6)** | 13.4 | 25.3 | | 18.2 | 29.1 | | 8.9 | 20.3 | | 0.001 |
| Before 2:00 PM | 4.6 | 14.8 | <0.001 | 6.3 | 17.6 | <0.001 | 3.1 | 11.4 | <0.001 | 0.046 |
| After 2:00 PM | 10.8 | 23.1 | | 15.0 | 27.1 | | 7.0 | 17.9 | | 0.002 |
| **Paired differences** | | | | | | | | | | |
| (1)–(3) | 26.5 | 35.8 | <0.001 | 29.5 | 36.8 | <0.001 | 23.7 | 34.8 | <0.001 | |
| (2)–(4) | 1.3 | 34.0 | 0.417 | 3.2 | 37.7 | 0.162 | −0.5 | 30.2 | 0.939 | |
| (5)–(6) | 4.5 | 28.0 | 0.004 | 5.1 | 30.9 | 0.026 | 3.8 | 25.1 | 0.056 | |

Notes:
PEC, physical education class; n, number of participants; SD, standard deviation.
$p^*$, p-value for the difference between Before and After 2:00 PM, between Before 2:00 PM in school day with (1) and without PEC (3), between After 2:00 PM in school day with (2) and without PEC (4), and between After 2:00 PM in school day (5) and whole weekend day (6) (Wilcoxon signed-rank test).
$p^\dagger$, p-value for the difference between boys and girls (Mann-Whitney U test).

recommendation to perform at least 60 minutes of average daily MVPA throughout the week (*Benavente-Marín et al., 2024*). Compliance with the MVPA recommendation shows a great variability among different studies, which can be explained by the true variability among populations, as well as by the use of different methodologies (*Van Hecke et al., 2016*). However, the differences found between compliance with the MVPA and the VIAs recommendation were observed under the same methodological conditions and in the same sample. This difference is mainly because the methods to operationalize the WHO recommendations on physical activity evaluate different physical activity patterns from each other.

In this sense, it could be understood that if the WHO incorporates two different recommendations on physical activity, it's to promote different health-related physical activity patterns. VIAs can include a certain proportion of VPA, but they do not necessarily have to be composed entirely of VPA. In fact, physical activity sessions carried out by children and adolescents with a significant vigorous intensity component (for example, participating in a football match) stand out for having a variable proportion of

**Table 5 Average percentage of days and daily segments with bouts of vigorous activities on different types of day, by sex and by extracurricular organized sport activities participation.**

| | | <3 h/week OSA | | | ≥3 h/week OSA | | | |
|---|---|---|---|---|---|---|---|---|
| n= | | Boys 75; Girls 91 | | | Boys 76; Girls 78 | | | |
| | | % | SD | $p^*$ | % | SD | $p^*$ | $p^\dagger$ |
| **Boys** | **School day with PEC** | | | | | | | |
| | Before 2:00 PM (1) | 32.4 | 35.9 | <0.001 | 33.6 | 37.3 | 0.163 | 0.847 |
| | After 2:00 PM (2) | 8.2 | 24.6 | | 43.4 | 39.9 | | <0.001 |
| | **School day without PEC** | | | | | | | |
| | Before 2:00 PM (3) | 2.0 | 10.2 | 0.001 | 5.3 | 14.7 | <0.001 | 0.080 |
| | After 2:00 PM (4) | 15.2 | 28.1 | | 30.8 | 33.7 | | 0.001 |
| | **Weekend day** | | | | | | | |
| | Whole day (5) | 9.0 | 20.0 | | 26.0 | 32.8 | | <0.001 |
| | **Paired differences** | | | | | | | |
| | (1)–(3) | 30.4 | 36.7 | <0.001 | 28.3 | 37.8 | <0.001 | |
| | (2)–(4) | −7.0 | 24.2 | 0.008 | 12.6 | 44.2 | 0.015 | |
| | (2)–(5) | −0.8 | 28.1 | 0.652 | 17.4 | 46.4 | 0.002 | |
| | (4)–(5) | 6.2 | 29.5 | 0.104 | 4.8 | 37.9 | 0.215 | |
| **Girls** | **School day with PEC** | | | | | | | |
| | Before 2:00 PM (1) | 29.1 | 38.0 | <0.001 | 21.6 | 32.4 | 0.610 | 0.223 |
| | After 2:00 PM (2) | 8.2 | 19.3 | | 18.8 | 28.3 | | 0.006 |
| | **School day without PEC** | | | | | | | |
| | Before 2:00 PM (3) | 0.4 | 3.5 | 0.003 | 1.3 | 8.0 | <0.001 | 0.465 |
| | After 2:00 PM (4) | 6.7 | 19.3 | | 20.2 | 28.6 | | <0.001 |
| | **Weekend day** | | | | | | | |
| | Whole day (5) | 8.0 | 19.3 | | 8.3 | 20.4 | | 0.917 |
| | **Paired differences** | | | | | | | |
| | (1)–(3) | 28.8 | 37.4 | <0.001 | 20.3 | 32.9 | <0.001 | |
| | (2)–(4) | 1.6 | 22.3 | 0.473 | −1.4 | 37.1 | 0.870 | |
| | (2)–(5) | 0.3 | 22.1 | 0.918 | 10.5 | 33.2 | 0.016 | |
| | (4)–(5) | −1.3 | 24.2 | 0.557 | 11.9 | 35.7 | 0.014 | |

Notes:
n, number of participants; h, hour; OSA, extracurricular organized sport activities; PEC, physical education classes; SD, standard deviation.

$p^*$, $p$-value for the difference between Before and After 2:00 PM, between Before 2:00 PM in school day with (1) and without PEC (3), between After 2:00 PM in school day with (2) and without PEC (4), between After 2:00 PM in school day with PEC (2) and whole weekend day (5), and between After 2:00 PM in school day without PEC (4) and whole weekend day (5) (Wilcoxon signed-rank test).

$p^\dagger$, $p$-value for the difference between children with less than three weekly hours and at least three weekly hours of OSA (Mann-Whitney U test).

physical activity at all intensity levels, but with a proportion of VPA greater than other activities that would not be considered vigorous sessions or activities. Therefore, the VIAs of the WHO recommendation and the VPA obtained through objective evaluation methods are not equivalent concepts.

The performance of VPA distributed throughout the day is beneficial for health, and its evaluation may be more interesting than that of daily MVPA (*Martinez-Gomez et al., 2010*;

**Table 6 Average percentage of children who meet the WHO recommendation of at least three weekly days with vigorous intensity activities.**

|  | n | n* | % | p |
|---|---|---|---|---|
| **All** | 353 | 84 | 23.8 | |
| Boys | 170 | 56 | 32.9 | <0.001 |
| Girls | 183 | 28 | 15.3 | |
| **All** | | | | |
| ≥3 h/week OSA | 154 | 54 | 35.1 | <0.001 |
| <3 h/week OSA | 166 | 21 | 12.7 | |
| **Boys** | | | | |
| ≥3 h/week OSA | 76 | 37 | 48.7 | <0.001 |
| <3 h/week OSA | 75 | 12 | 16.0 | |
| **Girls** | | | | |
| ≥3 h/week OSA | 78 | 17 | 21.8 | 0.032 |
| <3 h/week OSA | 91 | 9 | 9.9 | |

Notes:
$n$, number of total participants.
$n^*$, participants who meet the recommendation; h, hours; OSA, extracurricular organized sport activities; $p$, $p$-value for the difference between categories (Chi-squared test).

*Gralla et al., 2016*; *Füssenich et al., 2016*; *Schwarzfischer et al., 2017*; *Larsen et al., 2018*; *García-Hermoso et al., 2021*; *Gammon et al., 2022*; *Benavente-Marín et al., 2024*). However, daily VPA represents a pattern of physical activity performance different from the recommendation on VIAs and, in turn, similar to the performance of daily MVPA. For example, running for one or two minutes to avoid being late for class can contribute to the daily amount of VPA and MVPA, but it would not be enough to be considered as a physical activity session, as would participating in a football match for an hour. Therefore, the evaluation of VPA patterns concentrated in activity sessions is important for the study and differentiation of the two recommendations on physical activity published in the WHO Guidelines (*World Health Organization (WHO), 2020*).

Including different patterns of physical activity in habitual physical activity can improve cardiovascular health in children (*Stone et al., 2009*; *Dorsey, Herrin & Krumholz, 2011*; *Jenkins et al., 2017*; *Chinapaw et al., 2018*), highlighting the inclusion of VPA intervals or the distribution of physical activity in periods of at least 10 minutes (*Stone et al., 2009*; *Chinapaw et al., 2018*). In an intervention study conducted in children with overweight or obesity, of similar age and from the same region (Andalusia, Spain) as the participants in our study, it was possible to significantly reduce the cardiovascular risk of the participants by engaging in 3–5 weekly sessions of VIAs (*Migueles et al., 2023*). Therefore, replicating the VPA pattern recommended by the WHO through activities with a certain concentration of VPA is related to improvements in cardiovascular health.

Regarding the identification of VPA-Bouts (*i.e.*, intervals of at least 60 minutes with at least 16.7% in boys and 12.5% in girls of VPA), a large difference was observed between school days with and without physical education class during the school time (28.6% *vs* 2.1%; $p < 0.001$). This difference is mainly, if not exclusively, due to the presence of

physical education classes. However, the contribution of physical education classes to the physical activity levels could be conditioned by the methodology used by the teachers (*Huertas-Delgado et al., 2021*) or by the type of content (*Molina-García et al., 2016*). Even so, physical education classes contribute to the performance of more physical activity at all intensities, as well as to the compliance with the MVPA recommendation (*Meyer et al., 2013*; *Chen, Kim & Gao, 2014*; *Mooses et al., 2017*; *Calahorro-Cañada et al., 2017*; *Ikeda et al., 2022*; *Benavente-Marín et al., 2024*, *2023*). And as is evident, they also contribute to the performance of VIAs and, therefore, to the compliance with the WHO recommendation to incorporate this type of activities into the weekly physical activity.

No significant differences were observed in the proportion of segments After 2:00 PM with VPA-Bouts between days with and without physical education class (18.9% *vs* 17.6%; $p < 0.417$), while on the weekend a lower proportion of VPA-Bouts was observed (After 2:00 PM in school days: 17.8%, weekend day: 13.4%; $p = 0.004$). In other words, the distribution of extracurricular sports and physical activities carried out during school days seems to be evenly distributed. However, on the weekend there seems to be a lower proportion of this type of activities.

Another notable result for the out-of-school time was the similarity between the average daily duration of declared extracurricular organized sports activities and the average daily duration of VPA-Bouts in the out-of-school time (20.8 *vs* 17.2 daily minutes). However, when observing the results segmented by sex, the difference found in girls stands out (20.2 *vs* 11.0 daily minutes). It seems that part of the extracurricular sports and physical activities in which girls participate did not reach the minimum intensity levels to identify the performance of VIAs through accelerometry. Indeed, other studies show that girls tend to prefer individual sports with artistic connotations, while boys often engage more intensity activities as team contact sports (*Resaland et al., 2019*; *Peral-Suárez et al., 2020*). This highlights the importance of studying VIAs through objective methods of physical activity assessment, especially in girls.

When segmenting the participants between those who declared less than three or at least three weekly hours of organized extracurricular sports activities, it was observed that participating in more hours of extracurricular activities did not significantly influence the identification of VPA-Bouts during the school time, while it did so out-of-school time. When segmenting by sex, in boys, significant differences were found in VPA-Bouts in all segments of extracurricular time (*i.e.*, After 2:00 PM school days with and without physical education class, and weekends) between those who declared to perform at least three weekly hours of extracurricular sports activities and those who did not. However, in girls, no differences were found on weekends according to their participation in extracurricular sports activities. These results are in line with the preferences for participation in extracurricular sports activities of boys and girls (*Resaland et al., 2019*; *Peral-Suárez et al., 2020*). Team contact sports, which are more preferred by boys, usually have their competition periods on the weekend.

In girls, it was especially relevant to know the segmented results on school days with and without physical education class, and weekend days, since neither in the average weekly day nor in the average school day any differences were observed in the proportion of days

with VPA-Bouts between Before and After 2:00 PM (Before *vs* After, weekly day: 8.5% *vs* 12.1%, $p = 0.106$; school day: 10.7% *vs* 12.7%, $p = 0.405$). This highlights the impact of the school day with physical education class on the performance of physical activity throughout the week (*Meyer et al., 2013*; *Mooses et al., 2017*), especially in girls. It seems that girls are more dependent than boys on organized and mandatory physical activities (*i.e.*, physical education classes) to perform VIAs, as in extracurricular organized physical activities they seem to prefer those with a lower intensity component (*Resaland et al., 2019*; *Peral-Suárez et al., 2020*).

Most of the evidence that studies habitual physical activity segmented by sex or gender shows significant differences between boys and girls (*Martinez-Gomez et al., 2010*; *Laguna et al., 2013*; *Katzmarzyk et al., 2015*; *Füssenich et al., 2016*; *Corder et al., 2016*; *Schwarzfischer et al., 2017*; *Ferrer-Santos et al., 2021*). In this study, even though the proportion of VPA necessary to identify a VPA-Bout is lower in girls than in boys (*i.e.*, 12.5% *vs* 16.7% of VPA, respectively; *Benavente-Marín et al., 2023*), it was generally observed that female participants achieved a lower proportion of VPA-Bouts in all types of days compared to male participants. This resulted in the girls who met the VIAs recommendation being half that of the boys who complied (15.3% *vs* 32.9%). These differences are accentuated among boys and girls who reported performing at least three weekly hours of organized extracurricular sports activities (*Whiting et al., 2021*). This situation underscores that boys and girls appear to differ, not only in their preferences for participation in sports and activities (*Resaland et al., 2019*; *Peral-Suárez et al., 2020*), but these differences are also observed in the performance of habitual physical activity (*Benavente-Marín et al., 2024*), as well as in participation in physical activity sessions with a prominent vigorous component (*Molina-García et al., 2016*).

It seems that boys participate in more intense physical activity sessions and more frequently than girls. However, it is possible that girls achieve similar health levels to boys with less volume and intensity of physical activity (*Laguna et al., 2013*; *Katzmarzyk et al., 2015*; *Gralla et al., 2016*; *Schwarzfischer et al., 2017*). Therefore, it is advisable to continue investigating the relationship between different patterns of physical activity, health, and gender.

The main strength of this study was the use of accelerometry as a tool for measuring children's physical activity, avoiding biases inherent in other methods, such as self-reported questionnaires (*Sirard & Pate, 2001*; *Migueles et al., 2017*; *Gao et al., 2021*). The design of a protocol with the GENEActiv accelerometer placed on the wrist optimized adherence in our study (*Fairclough et al., 2016*). In addition, the GENEActiv accelerometer allowed us to obtain raw data without any prior filtering hidden under license, which can underestimate the results of high-intensity physical activity, especially in children (*Rowlands et al., 2016*; *Arvidsson, Fridolfsson & Börjesson, 2019*).

Despite the strengths of this study, it also has limitations. The main limitation was the reduction in the evaluated sample, compared to the planned one, due to the closure of schools and the home confinement caused by the COVID-19 pandemic. Even so, the evaluated sample exceeds the minimum size necessary to meet the objectives. However, participants could not be evaluated during the spring, with more daylight hours and better

weather conditions for physical activity (*Remmers et al., 2017*; *Turrisi et al., 2021*). On the other hand, there are inherent limitations to the study of physical activity through accelerometry when comparing results from different studies, such as the accelerometer model, anatomical location, methodology for processing raw data, with emphasis on the duration of epochs, or the cut-off points selected to determine the intensity of physical activity (*Baquet et al., 2007*; *Rowlands et al., 2016*; *Aadland et al., 2018*; *Arvidsson, Fridolfsson & Börjesson, 2019*; *Llorente-Cantarero et al., 2021*). In addition, accelerometry does not adequately capture the intensity caused by physical activities such as cycling or resistance training with overload. Therefore, the combination of accelerometry with heart rate measurement can offer more accurate results (*Van Camp, Batchelder & Irwin Helvey, 2022*). Participants were restricted to children aged 8–9 from a specific region of Spain, so the findings may not be generalizable to younger children or older youth, as well as to children from other regions or countries. Finally, the observational nature of this cross-sectional study excludes any cause-and-effect association between physical activity or compliance with recommendations and sex, overweight, or performance in physical education classes or extracurricular activities.

## CONCLUSIONS

One in every four children aged 8–9 in Andalusia (Spain) met the WHO recommendation to "incorporate vigorous intensity activities at least three days a week" evaluated by accelerometry. Compliance was higher on days with physical education classes, as well as among children who reported participating in organized extracurricular sports activities. Therefore, physical education classes and extracurricular organized sports activities contributed to improving compliance with the WHO's recommendation on vigorous intensity activities.

## ACKNOWLEDGEMENTS

We thank the staff, pupils, parents, schools, and municipalities for their participation, enthusiasm, and support.

### Funding

This research was funded by Proyecto I+D+I Programa Operativo FEDER Andalucía 2014–2020 (UMA18-FEDERJA-114), and the University of Málaga. The funders had no role in study design, data collection and analysis, decision to publish, or preparation of the manuscript.

### Grant Disclosures

The following grant information was disclosed by the authors:
Proyecto I+D+I Programa Operativo FEDER Andalucía 2014–2020; UMA18-FEDERJA-114.
University of Málaga.

## Competing Interests

Napoleón Pérez-Farinós and Julia Wärnberg are Academic Editors for PeerJ.

## Author Contributions

- Juan Carlos Benavente-Marín conceived and designed the experiments, performed the experiments, analyzed the data, prepared figures and/or tables, authored or reviewed drafts of the article, and approved the final draft.
- Francisco Javier Barón-López conceived and designed the experiments, analyzed the data, prepared figures and/or tables, authored or reviewed drafts of the article, and approved the final draft.
- Begoña Gil Barcenilla performed the experiments, authored or reviewed drafts of the article, and approved the final draft.
- Guadalupe Longo Abril performed the experiments, authored or reviewed drafts of the article, and approved the final draft.
- José M. Rumbao Aguirre performed the experiments, authored or reviewed drafts of the article, and approved the final draft.
- Napoleón Pérez-Farinós conceived and designed the experiments, analyzed the data, prepared figures and/or tables, authored or reviewed drafts of the article, and approved the final draft.
- Julia Wärnberg conceived and designed the experiments, analyzed the data, prepared figures and/or tables, authored or reviewed drafts of the article, and approved the final draft.

## Human Ethics

The following information was supplied relating to ethical approvals (*i.e.*, approving body and any reference numbers):

The study was conducted in accordance with the Declaration of Helsinki, and approved by the research ethics committee CEI-Costa del Sol and the Portal de Ética de la Investigación Biomédica de Andalucía-PEIBA, the 26th of September 2019, with the reference number 0114-2019.

## Data Availability

The data that support the findings of this study are restricted because the subjects are vulnerable children in schools. Sensitive data such as weight and height cannot be published even after deidentification given the small population that these subjects are drawn from. Permission to access the data can be requested from the ethics commission https://www.juntadeandalucia.es/salud/portaldeetica, email: portaldeetica.csalud@juntadeandalucia.es.

## Supplemental Information

Supplemental information for this article can be found online at http://dx.doi.org/10.7717/peerj.16815#supplemental-information.

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
