# Peer review of "Adherence to the WHO recommendation of three weekly days of vigorous intensity activities in children: an accelerometry study of vigorous physical activity bouts"

_PeerJ, doi:10.7717/peerj.16815_

## Round 0.1 · original submission · Minor Revisions

Dear Authors:

Please, response to reviewers´ comments. Manuscript is a good paper to be published on PeerJ but it needs to be improved.

Reviewers have suggested that you cite specific references. You are welcome to add it/them if you believe they are relevant. However, you are not required to include these citations, and if you do not include them, this will not influence my decision.

Dr. Manuel Jiménez

Reviewer 1 ·

Basic reporting

Thank you for the opportunity to review the manuscript entitled "Adherence to the WHO Recommendation of Three Weekly Days of Vigorous Intensity Activities in Children: An Accelerometry Study of Vigorous Physical Activity Bouts". The main aim of this study was to identify vigorous intensity activities (VIA) through bouts of vigorous physical activity evaluated with accelerometry and, with this, to know the compliance with the recommendation on VIA. I would like to congratulate the authors for doing an analysis focused on vigorous activities. As we know, in the literature, it is common for moderate and vigorous intensity activities to be analyzed together and not separately. The manuscript is well-written, easily comprehensible, and incorporates relevant and suitable references. The structure is appropriate, providing relevant introduction and background.

While the manuscript is suitable for publication, I recommend a few minor corrections:

1.- Tables:
- Enhance the titles of tables, particularly Table 3 and 4. Simplify the titles, for instance, changing "In the whole day, in the school time, and after school time" to "Daily Segments."
- In Table 2 and 3 footnotes, replace "chi-square" with "chi-square test."

2.- Ethics:
- Ensure that the ethics paragraph references the Declaration of Helsinki from 2013 (doi:10.1001/jama.2013.281053).
- Properly reference the adherence to STROBE guidelines (DOI: 10.4103/sja.SJA_543_18).

3.- Throughout the text, consider using "extracurricular organized sports activities" instead of "organized sports outside of school."

4.- Discussion section: Many studies indicate the significant role of physical education in the physical activity levels of young people. However, the contribution of this subject to the physical activity levels is highly conditioned by the methodology used by the teacher (Huertas-Delgado et al., 2021) or by the type of content (Molina-García et al., 2016). For example, content related to body expression corresponds to low levels of physical activity, while content related to physical fitness contributes to a greater extent to physical activity levels. I recommend that the authors indicate this in the Discussion section.

Experimental design

The study aimed to identify vigorous intensity activity sessions (e.g., sports activities, active games, or physical activity classes) using previously defined VPA bouts measured with accelerometers, and presenting compliance with WHO recommendations for at least 3 days of sessions per week. The research question is pertinent, addressing a knowledge gap in a poorly studied area. The manuscript is interesting, well-written, and features a robust methodology.

Validity of the findings

Accessing the dataset has confirmed its statistical soundness. The conclusions are appropriately drawn, aligning with the original question and supported by results. The definition of the bout used is defined in a previous article.

The main concern lies in the lack of standardized cutoff points using raw accelerometer data. However, the author's use of cutoff points defined by Hildebrand et al. and the clear methodological description make the novel approach reproducible by other researchers.

Reviewer 2 ·

Basic reporting

The aim of this study is to present compliance with a recommendation of engaging in vigorous activities at least 3 days of sessions per week, in a representative sample of children 8-9 years old, from Andalusia, Spain.
The novelty is that the compliance is measured with accelerometers. It is an interesting manuscript, well written and with a robust methodology.
The language used is clear and professional. However, please read carefully through the paper again focusing on simplifying some expressions.
Below are some suggestions of improvements.

Experimental design

-Within the methodology section:
The following question is used: “Is your child a member of one or more sports or dance clubs (e.g., football, athletics, hockey, swimming, tennis, basketball, judo, taekwondo, gymnastics, ballet, physical training, ballroom dancing, etc.) or does he/she take classes in them?” The possible answers were “Yes” or “No”. In case of an affirmative answer, the following question was asked: “In a normal week (including the weekend), how many hours does your child spend in these types of sports and physical activities?”
There are no details of how this question was designed and I would suggest to include a brief background of the choice of this question.

In the ethics statement I recommend the Declaration of Helsinki to be properly cited, and some more details of how confidentiality is guaranteed, and if the individual results were informed to the participants or parents.
Were the schools given any kind of feedback of the physical activity levels of the evaluated class?

-Within the discussion section:
In this study, by design and previous work, you consider that there is a gender difference in the need of VPA for health, by using different definitions of the VPA bouts for girls and boys that are used to detect the sessions of vigorous activities. However, using these lower cutoff points in girls they still engage in less active extracurricular activities. Below I recommend you to include some references that could add to the discussion of the preferences of sports by gender.

Peral-Suárez Á, Cuadrado-Soto E, Perea JM, Navia B, López-Sobaler AM, Ortega RM. 2020. Physical activity practice and sports preferences in a group of Spanish schoolchildren depending on sex and parental care: a gender perspective. BMC Pediatrics 20:337. DOI: 10.1186/s12887-020-02229-z.
Resaland GK, Aadland E, Andersen JR, Bartholomew JB, Anderssen SA, Moe VF. 2019. Physical activity preferences of 10-year-old children and identified activities with positive and negative associations to cardiorespiratory fitness. Acta Paediatrica 108:354–360. DOI: 10.1111/apa.14487.

Validity of the findings

The article is original research and is well suited to the journal. The research question is well defined, relevant and add evidence to the topic.
The articles technical and ethical standard is high and the study is reproducible.

Additional comments

None

---

## Round 0.2 · accepted · Accept

Dear Co-Authors:

It is a pleasure to inform you that the manuscript entitled: "Adherence to the WHO recommendation of three weekly days of vigorous intensity activities in children: an accelerometry study of vigorous physical activity bouts" has been accepted for publication in the journal PeerJ. I want to thank you for trusting in this journal and I want to congratulate you for the great work done; it is a really nice paper.

Have a Happy New Year and congratulations.

Dr. Manuel Jimenez

Reviewer 1 ·

Basic reporting

No comment. Thank you for incorporating the previous comments. No further changes suggested to this paper.

Experimental design

No comment.

Validity of the findings

No comment.

Reviewer 2 ·

Basic reporting

I accept the changes made by the authors.

Experimental design

I accept the changes made by the authors.

Validity of the findings

No comments

Additional comments

No comments